

# Lightweight graph convolutional network with multi-attention mechanisms for intelligent action recognition in online physical education

Yuhao You

Department of Physical Education, Tongji University, Shanghai, China

## ABSTRACT

The rise of online physical education in higher education has improved accessibility but presents challenges in recognizing complex movements and delivering individualized feedback. Existing action recognition models are often computationally intensive and struggle to generalize across diverse skeletal patterns. To address this, we propose a lightweight graph convolutional network (GCN) that integrates an improved Ghost module with multi-attention mechanisms, including a global attention mechanism (GAM) and a channel attention mechanism (CAM), to enhance spatial and temporal feature extraction. The model is trained end-to-end on 3D skeleton sequences and optimized for real-time efficiency. The computational cost is evaluated in terms of giga floating-point operations (GFLOPs), with the proposed model requiring only 6.2 GFLOPs per inference, over 60% less than the baseline ST-GCN. Experimental results on the NTU60RGB+D dataset demonstrate that the model achieves 90.8% accuracy in cross-subject and 96.8% in cross-view settings. These findings highlight the model's effectiveness in balancing accuracy and efficiency, with promising applications in online physical education, rehabilitation monitoring, elderly movement analysis, and VR-based interfaces.

# INTRODUCTION

The continuous advancement of modern technology, particularly artificial intelligence (AI), has brought transformative changes to various fields, including education. Physical education (PE), as an essential component of holistic education, traditionally relies on face-to-face teaching and real-time guidance provided by professional instructors. This is especially critical for anaerobic strength training, which requires accurate demonstrations, precise feedback, and personalized adjustments (*Syaukani, Hashim & Subekti, 2023*). However, current fitness management practices in educational institutions often fail to adequately address these requirements. For instance, many schools rely on running-based management software to record students' physical activity. While effective in managing attendance and promoting aerobic exercises, such systems are limited to basic fitness tracking and fail to incorporate anaerobic training due to its inherent complexity and the

Corresponding author
Yuhao You, youyuhao@tongji.edu.cn

need for professional supervision (*Jiang, 2024*). This lack of comprehensive guidance in fitness education underscores the need for innovative and intelligent solutions to bridge this gap and enhance the effectiveness of physical education programs.

Skeleton-based human action recognition has emerged as a promising approach to address these challenges (*Wang & Yan, 2023*; *Shi, 2023*). Compared to traditional red-green-blue (RGB)-based methods, skeleton-based AR provides a compact and robust representation of human movements using 3D skeleton data. By capturing the positions and motions of key skeletal joints, this method avoids the high computational costs and environmental sensitivity associated with RGB-based approaches. Additionally, skeleton-based AR demonstrates strong robustness and accuracy, making it suitable for real-world applications in education, intelligent healthcare, video understanding, and human-computer interaction (*Liu et al., 2025*; *Li et al., 2024*; *Leus et al., 2023*). These advantages have led to the widespread adoption of skeleton sequences in intelligent systems, where human action can be effectively represented and analyzed. In previous studies, skeleton-based AR methods typically relied on handcrafted feature extraction techniques. Joint points of the human body were often treated as independent features, with spatial-temporal correlations modeled through manual design (*Wang et al., 2024*). For example, *Sun et al. (2022)* proposed Eigen Joints, which encode static poses, motion trajectories, and offsets for action recognition using a naive Bayes nearest neighbor (NBNN) classifier. While effective in some scenarios, such methods tend to neglect the interdependencies between skeletal joints, resulting in limited accuracy and scalability. Moreover, the complexity of handcrafted features makes these approaches less practical for large-scale applications, prompting researchers to turn to deep learning methods.

Recent advances in deep learning have revolutionized skeleton-based augmented reality (AR) by enabling the automatic extraction and modeling of complex joint relationships. Three main frameworks have been widely adopted in this domain: convolutional neural networks (CNNs) (*Ali et al., 2023*), recurrent neural networks (RNNs) (*Usmani, Siddiqui & Islam, 2023*), and graph convolutional networks (GCNs) (*Cui, Ding & Chen, 2024*). CNN-based approaches, such as the method by *Huang et al. (2025)*, map joint points into 3D coordinate spaces and separately encode spatial and temporal information to extract deep features using 3D CNNs. While this approach achieves high recognition accuracy, it suffers from excessive computational complexity due to the large number of parameters involved. RNN-based methods, such as the cascaded RNN proposed by *Du, Wang & Wang (2015)*, segment the skeleton into five body parts based on human anatomy, feed them into independent sub-networks, and fuse their outputs layer by layer to classify actions. Although effective in capturing temporal dependencies, this method also faces optimization challenges due to its computational demands.

In contrast, GCNs have gained popularity for their ability to model both spatial and temporal relationships in skeletal data. By leveraging the structural information of key joints and integrating temporal sequences, GCNs achieve superior performance in action recognition tasks (*Ahmad et al., 2021*). In addition to modeling capabilities, computational cost is a critical factor in skeleton-based AR, especially for real-time applications. CNN-based methods, while accurate, often exceed 50–100 giga floating point operations

(GFLOPs) due to the use of dense 3D convolutions and high parameter counts (*Huang et al., 2025*). RNN-based models typically require fewer FLOPs than 3D CNNs but still incur significant computational overhead during sequential processing, often reaching 20–30 GFLOPs depending on the network depth (*Du, Wang & Wang, 2015*).

In contrast, GCNs strike a better balance between recognition accuracy and efficiency. For instance, the baseline ST-GCN model operates at 16.2 GFLOPs, with most costs concentrated in temporal graph convolutions. These comparisons highlight GCNs as a more computationally efficient foundation for designing lightweight and scalable frameworks for action recognition.

Despite these advances, existing skeleton-based AR methods still face several challenges. First, the computational cost of these models remains high, making them difficult to deploy in resource-constrained environments. Second, the accuracy of action recognition is often limited by the inability to capture subtle joint relationships and contextual information fully. To address these issues, this study proposes a lightweight Ghost GCN model designed to improve the efficiency and accuracy of skeleton-based AR. The proposed model incorporates two key innovations: (1) the integration of an improved Ghost module (*Han et al., 2020*), which reduces computational complexity and creates a lightweight GCN architecture, and (2) the design of two attention mechanisms, a global attention mechanism (GAM) and a channel attention mechanism (CAM), to enhance the model's ability to capture critical spatial and temporal features.

The remainder of this article is organized as follows: "Related Works" reviews recent literature on skeleton-based action recognition and lightweight network architectures. "Implementation of Online P.E. Classes Based on Lightweight GCN" details the proposed lightweight graph convolutional neural network (GCN) model, including its architectural components, such as the Ghost module, graph attention mechanisms, and channel attention modules. "Experiment and Analysis" presents experimental settings, quantitative evaluations, and ablation studies based on the NTU60RGB+D dataset. "Conclusion" concludes the article by summarizing key contributions and outlining future research directions.

## RELATED WORKS

Human body joint point features can be accurately captured using human pose estimation algorithms or high-precision depth cameras. By connecting these key points, the human skeleton can be represented as a graph structure, which is particularly effective for action recognition tasks. *Yan, Xiong & Lin (2018)* proposed a spatio-temporal graph convolutional network (ST-GCN), which extends traditional graph convolutional networks to the spatiotemporal domain. The core innovation of ST-GCN lies in constructing a spatiotemporal graph structure from input skeletal key points. This graph not only preserves the spatial relationships among the joints but also incorporates the temporal trajectories of key points by modeling them as time-series edges. This spatiotemporal representation significantly enhances the network's ability to capture dynamic action features, improving robustness and feature performance in human action recognition tasks.

Building upon ST-GCN, *Liu et al. (2023)* introduced the two-stream adaptive graph convolutional network (2s-AGCN). This model employs a self-learning adjacency matrix strategy, enabling it to adapt the graph structure during training dynamically. This self-adaptive mechanism enhances the extraction of spatial features, allowing the network to capture complex joint dependencies in human motion more effectively. Similarly, *Zhu & Ren (2021)* developed the action-structural graph convolutional network (AS-GCN), which introduces an innovative mechanism to extract two distinct types of graph links: action links and structural links. These links are derived from the original joint coordinates and provide a more comprehensive representation of human actions by simultaneously modeling motion-related and structural features.

Further advancements in this field include the work of researchers who have explored hybrid architectures combining GCNs with other deep-learning techniques. For instance, *Zhang et al. (2024)* proposed the AGC-LSTM model, which integrates an attention-enhanced GCN with a LSTM network. This hybrid architecture utilizes an attention mechanism to emphasize the features of key skeletal points while leveraging the LSTM to enhance the modeling of high-level spatiotemporal semantic features. The combination of these two components enables the AGC-LSTM to effectively capture both spatial dependencies and temporal dynamics, resulting in improved action recognition performance. In another approach, *He et al. (2024)* introduced the directed graph neural network (DGNN), which employs a directed acyclic graph (DAG) to represent the human skeletal structure. Unlike traditional undirected graph models, the DGNN dynamically adjusts its topology during the training process to better adapt to the requirements of action recognition tasks. By incorporating motion information and spatial information from skeleton sequences, this model further improves the performance of two-stream frameworks. The directed nature of the graph enables a more accurate representation of joint relationships, enhancing the overall effectiveness of the model.

In recent years, lightweight network architectures have gained considerable attention in the field of action recognition. These networks aim to reduce the overall size and computational complexity of models while maintaining strong feature extraction capabilities (*Cao et al., 2024*). Lightweight networks can be categorized into three primary strategies: model lightening, network slimming, and direct design of lightweight architectures. The lightning strategy focuses on reducing the number of parameters in the network. For instance, *Hong et al. (2024)* proposed an 8-bit integer fixed-point representation to eliminate redundant parameters, thereby reducing the overall model size without compromising accuracy. Similarly, *Cui, Li & Zhang (2021)* introduced a quantization method for dense weight matrices to achieve significant compression of network parameters. Compared to conventional networks, lightweight architectures require fewer parameters and achieve lower Floating-point Operations Per Second (FLOPs), making them more efficient. This efficiency makes lightweight networks particularly suitable for deployment in embedded devices and mobile terminals. For example, lightweight network designs enable real-time applications in environments with limited computational resources, such as wearable devices or smartphones, without compromising recognition accuracy (*Qiu, 2024*; *Ren et al., 2024*). By maintaining a balance

between computational efficiency and feature extraction performance, lightweight networks are becoming an increasingly popular choice for practical applications in human action recognition.

# IMPLEMENTATION OF ONLINE P.E. CLASSES BASED ON LIGHTWEIGHT GCN

To effectively recognize human skeletal actions in online physical education scenarios, the proposed model introduces a lightweight spatiotemporal GCN enhanced with multi-attention mechanisms. The architecture is built upon the ST-GCN backbone, with the spatial graph convolution replaced by an improved Ghost module to reduce computational costs. A ResNet block and dropout are incorporated to stabilize training and mitigate overfitting. To enhance the model's feature representation capabilities, a data-driven graph attention mechanism (DDGM) and a GAM are integrated into the spatial graph convolution layer, allowing for adaptive learning of skeletal topology. Additionally, a CAM is embedded to refine the extracted features in the channel dimension.

The network structure is designed based on the ST-GCN structure (*Yan, Xiong & Lin, 2018*). The ST-GCN network comprises a total of 10 layers, with each layer consisting of a spatial GCN (S-GCN) and a temporal GCN (T-GCN). The S-GCN is the core part of the designed structure. The structure of each layer in ST-GCN is illustrated in Fig. 1.

Based on the ST-GCN structure, this article uses the ghost convolution model in Ghost Net to reduce the amounts of parameters. In detail, the spatial graph convolution is replaced by the Ghost convolution, and a batch normalization (BN) and rectified linear unit (ReLU) are added to speed up the training process; therefore, the improved lightweight graph convolution model (LGCM) includes a ghost model, dropout, batch normalization (BN), and a rectified linear unit (ReLU) activation function. Among them, the dropout probability is 0.5. Additionally, the Resnet model is added to maintain a stable training process.

To enhance the feature extraction capability, we incorporate a graph attention module into the spatial graph convolutional layer, building upon the lightweight GCN module. It mainly consists of two parts: the DDGM and the GAM.

To enhance the recognition effectiveness and accuracy of motion recognition, a CAM is added to the network. In summary, the model of the lightweight spatiotemporal graph convolutional network, based on multi-attention, designed here is illustrated in Fig. 2.

## Graph convolutional networks

GAN methods mainly consist of spectral-based methods and spatial-based methods. The data flow in GCN, specifically at the spectrum-based graph convolution network, can be divided into three parts. First, the input data is transformed from the spatial domain into the frequency domain. Second, it is filtered. Third, it is restored to the spatial domain where the original graph signal is located. In this way, the features can be extracted entirely. However, the disadvantages of them are low flexibility, poor universality, and low

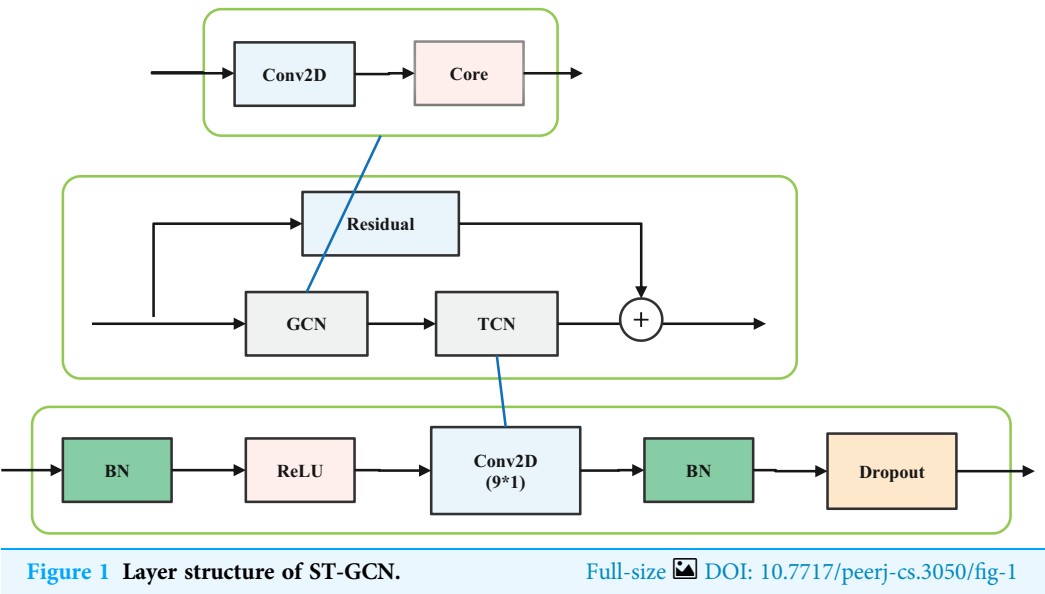

**Figure 1** Layer structure of ST-GCN. 

operating efficiency. Therefore, spatial-based methods can reduce complexity, enhance generalization ability, and improve operational efficiency.

The spatial domain graph convolution method is the mainstream method in the field of skeleton motion recognition. The skeleton data is described as N nodes, and the spatiotemporal graph of the T frame is $G = (V, E)$. The skeleton data coordinates of human actions can be expressed as $X \in \mathbb{R}^{N \times T \times d}$, however, d, here in the formula, is marked as the joint points size.

The model based on the GCN includes two convolution operations: SGC and TGC. As for SGC, the neighborhood of a node is bounded as an adjacency matrix $A \in \{0, 1\}^{N \times N}$. To better illustrate the SGC, the adjacency matrix is divided into three parts: centripetal points, eigen points, and centrifugal points. For each frame, $F \in \mathbb{R}^{N \times C}$ represents the input feature and $F' \in \mathbb{R}^{N \times C'}$ depicts the output feature; C and C' represent the dimensions of the input and output features, respectively. Formula (1) is the corresponding relationship between calculations in GCN.

$$F' = \sum_{p \in P} \bar{A}_p F W_p. \tag{1}$$

Among them, p = {eigen point, centripetal point, centrifugal point} represents a space partition; $\bar{A}_p$ is the normalized adjacency matrix, defined as Formula (2).

$$\bar{A}_p = \Lambda_p^{-\frac{1}{2}} A_p \Lambda_p^{-\frac{1}{2}} \in \mathbb{R}^{N \times N} \tag{2}$$

where $\Lambda_p^{ii} = \sum_j \left( A_p^{ij} \right) + \alpha$, to avoid empty lines, $\alpha$ is set to 0.001. The weights of the $1 \times 1$ convolutions for each partition are $W_p \in \mathbb{R}^{1 \times 1 \times C \times C'}$. For example, ST-GCN (*Yan, Xiong & Lin, 2018*) requires 16.2 GFLOPs to recognize an action example, with the spatial graph convolution consuming 4.0 GFLOPs and the temporal graph convolution consuming 12.2 GFLOPs. Some algorithms related to ST-GCN even require the consumption of

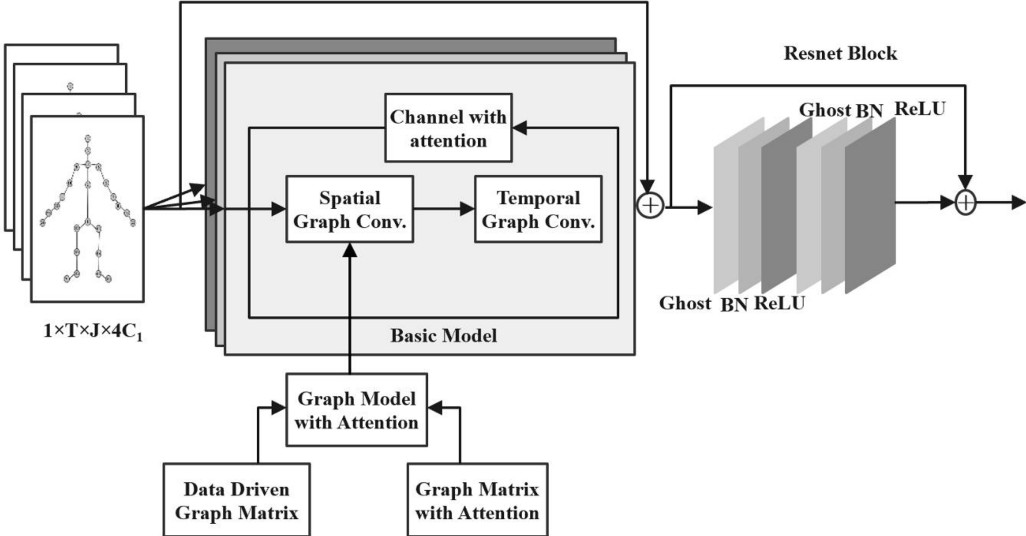

**Figure 2 Lightweight graph neural network model based on attention mechanism.**

100 GFLOPs (*Shi et al., 2019*; *Li et al., 2019*; *Wang et al., 2022*). Both the temporal graph structure and the spatial graph structure are predefined, although *Liu et al. (2023)* use a learnable adjacency matrix; however, it is still limited by the conventional graph convolution architecture.

## Improved ghost model

Limited by memory and computing power, it is challenging and literally impossible to construct neural networks on some embedded devices. For example, given input data, when the input size is two dimensions, h and w, the label c represents the channel numbers of the human skeleton data. Moreover, the output data is explicitly represented by a matrix of size h′ and w′. Among all the labels in the formula, f is used here to represent the abstraction of a filter. Furthermore, the kernel size of the convolution operation can be denoted by k. Considering the size of filters and kernels, which are often extremely large, such as 256 filters and channels, whose number is twice that of filters. Therefore, we can conclude that the FLOPs options can be in the hundreds of thousands or more. The conventional convolution operation is shown in Fig. 3.

A trained DNN often contains a considerable number of redundant feature maps. The high similarity among these maps is detrimental to the training process, as it leads to unnecessary computational overhead. Some feature maps are nearly identical, making it redundant to maintain such a large number of parameters and FLOPs. It is therefore hypothesized that there exists an intrinsic set of feature maps Y, which are generated by the initial convolution. The corresponding formulation is given in Eq. (3):

$$Y' = X * f' \tag{3}$$

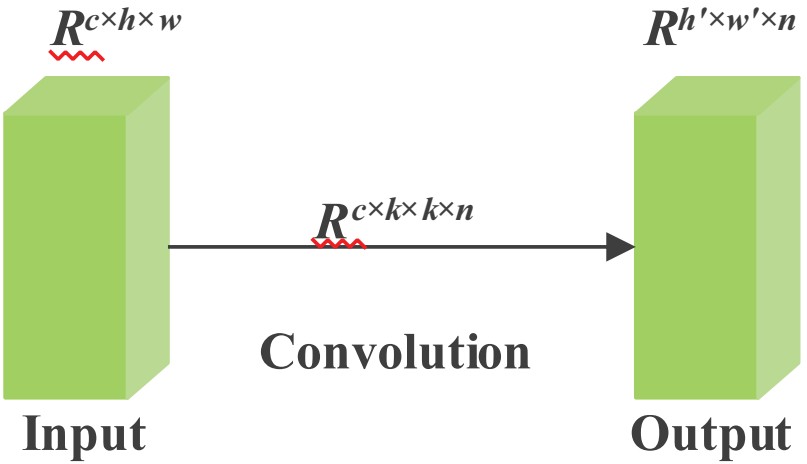

**Figure 3  Conventional convolution operation.**

$f' \in R^{c \times k \times k \times m}$ is still the name of a filter, and the limitation of the formula is that m is less than n. Where c is the number of input channels, k × k is the kernel size, and m is the number of intrinsic feature maps generated by the initial convolution layer. Others, such as the convolution kernel, stride, and spatial size, maintain similarities to regular convolution. Y = [y11, y12, ..., yms] will be the output data of the ghost model, as shown in Fig. 4.

Improved ghost models include identity maps and linear operations. By that, calculations of the designed effective ghost model are shown in Eq. (4). Replacing the ordinary convolution operation with the ghost module can reduce the number of parameters through s.

$$r_c = \frac{n \cdot c \cdot k \cdot k}{\frac{n}{s} \cdot c \cdot k \cdot k + (s-1) \cdot \frac{n}{s} \cdot d \cdot d} \approx \frac{s \cdot c}{s + c - 1} \approx s. \tag{4}$$

**Attention mechanism design**

To enhance the model's accuracy in recognizing human skeleton data and its feature extraction capabilities, attention modules are added to the spatial graph convolution operation and the channel, respectively.

*Graph attention mechanism model*

The introduction of the graph attention mechanism in the spatial graph convolutional (SGC) layer enables the model to optimize the connectivity graph while learning network parameters, thereby enhancing its action recognition ability. With the addition of the graph attention (GA) mechanism, the SGC formula is shown as follows:

$$f_{out} = \sum_{k}^{Kv} w_k(f_{in}(A_k' + B_k)). \tag{5}$$

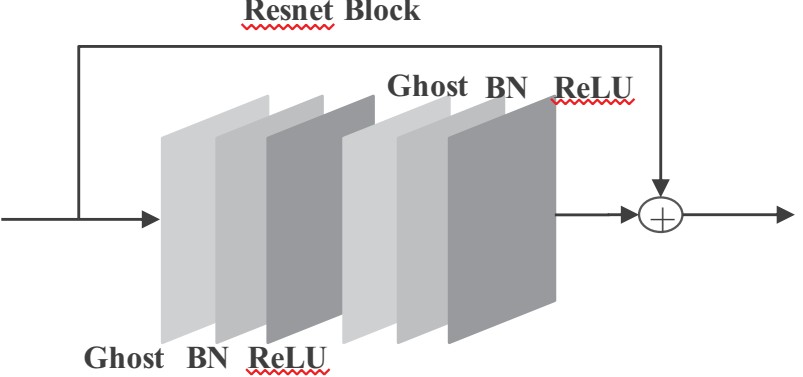

**Figure 4 Improved Ghost model.**

Matrix A′ denotes a data-driven graph (DDG), which is initially unparameterized in the GNN and subsequently updated during the training process. A′ leverages the original physical connections as a foundation while optimizing the topology of the connection graph and adaptively updating the edge weights. As A′ is entirely derived from training data, the resulting structure can be flexibly adapted to various types of skeleton-based action data.

In addition, the graph attention matrix B serves to capture fine-grained motion characteristics for each sample, thereby enhancing both the expressiveness and robustness of the network. The formulation of B is provided in Eq. (6).

$$S = sigmoid(W_2 ReLU(W_1 Z)). \tag{6}$$

For an input $f(n_{ti})$ (feature of a node), two convolutional layers firstly map $f(n_{ti})$ to K vectors and Q vectors. W represents the weight matrix, and the inner product is obtained for the above Q and K vectors. The results of the inner product are referred to as the similarity between nodes. Since the inner product ranges from 0 to 1, the vector values are normalized by using the softmax function.

Matrix B, which is the graph attention matrix, is obtained through various motions. It can fittingly study the weights between body joints in distinguishable motions. This DDG enhances the versatility and robustness of the net structure, enabling it to identify anaerobic motions more effectively.

Both the data-driven graph matrix A′ and the attention matrix B are learned in an end-to-end manner during training. Specifically, A′ is initialized based on physical skeletal connectivity but is not fixed—it is updated dynamically at each training epoch through backpropagation. This allows the graph topology to adapt progressively to diverse motion patterns in the training data. Meanwhile, the attention matrix B, computed *via* learned query-key mappings, is recalculated for each input sample and frame, ensuring that joint-level dependencies are adaptively weighted across different actions.

*Channel attention mechanism*

In each layer of the SGC network, the spatial features are preliminarily extracted *via* the graph attention module. The formulations of matrices Z and S are provided in Eqs. (7) and (8), respectively.

$$Z = \frac{1}{H * W} \sum_{i=1}^{H} \sum_{j=1}^{W} m_c(i,j) \tag{7}$$

$$S = sigmoid(W_2 ReLU(W_1 Z)). \tag{8}$$

The CAM is appended after the spatial graph convolution to refine feature weighting. Specifically, its operation involves three steps: First, the output of each layer is compressed to obtain matrix Z; second, Z is passed through a fully connected layer to yield matrix S; finally, S is element-wise multiplied with the original input feature map and combined with a residual connection to produce the final output **f**.

## EXPERIMENT AND ANALYSIS

The experimental environment in this article is set up as follows: 64-bit Ubuntu 18.04 operating system, Intel Xeon CPUE5-2678v3 @ 2.50 GHz, 12 GB memory, graphics card RTX 2080 Ti, CUDA 10.0.130, cuDNN 7.5, PyTorch 1.4, and Python 3.6 software platforms.

### Experiments datasets

To realize the reform and research on online teaching courses in physical education at colleges and universities, the data used here is the human skeleton dataset, including the NTU60RGB+D, which contains data on the human skeleton under various actions.

The NTU60RGB+D dataset was proposed by Nanyang Technological University and captured simultaneously by three Microsoft Kinect v2 cameras, comprising 56,880 action clips, 60 action categories, 17 camera placement combinations, and 40 actors participating in the dataset collection. Figure 5 illustrates the specific sampling point distribution, with a total of 25 sample joint points.

### Experiments of lightweight GCN with improved Ghost model

The improved lightweight GCN with an enhanced Ghost model is tested using the NTU60RGB-D dataset. What's more, the proposed improved Ghost module is compared with the most advanced skeleton-based action recognition method in terms of accuracy and parameter quantity. The experimental results are presented in Table 1, where CS represents Cross-Sub, CV represents Cross-View, and GFLOPs represent floating-point operations per second.

To further validate the lightweight characteristics of the proposed Ghost-GCN model, we compare its parameter count and model size with those of several representative skeleton-based action recognition methods. As shown in Table 1, the proposed model achieves state-of-the-art accuracy (90.8% CS/96.8% CV) while maintaining the lowest parameter count (1.9M) and smallest model size (~7.6 MB). Compared with ST-GCN, which requires 3.1 million parameters and 16.2 GFLOPs, our method reduces the

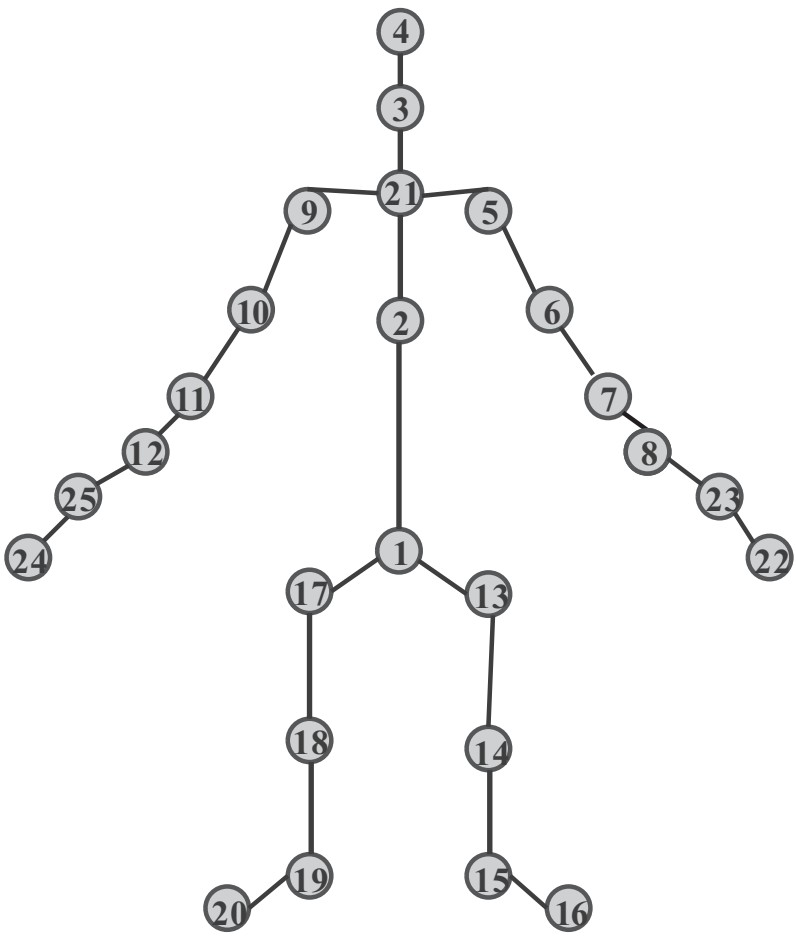

**Figure 5 Joint point labels of NTU60 RGB-D dataset.**

**Table 1 Comparison of experimental results on NTU60 RGB+G dataset.**

| Methods | CS/% | CV/% | GFLOPs |
|---|---|---|---|
| ST-GCN (*Yan, Xiong & Lin, 2018*) | 81.7 | 88.6 | — |
| 2s AS-GCN (*Liu et al., 2023*) | 86.9 | 94.0 | 27.0 |
| 2s AGCN (*Li et al., 2019*) | 88.3 | 95.0 | 35.8 |
| Improved Ghost GCN | 90.8 | 96.8 | 9.4 |

parameter size by approximately 38.7% and the computational cost by over 60% while improving accuracy by more than 9%. Similarly, it outperforms complex architectures like 2s-AGCN and AGC-LSTM not only in accuracy but also in compactness, making it highly suitable for deployment in real-time, resource-constrained scenarios such as mobile devices and online physical education platforms. These results underscore the model's effectiveness in striking a balance between recognition performance and computational efficiency.

This article employs two types of evaluation criteria in this dataset: (1) Cross-actor (Cross-Sub), which refers to actions collected from different actors. The actions demonstrated by actors identified 1~38 are used for training. And the actions shown by actors whose identities are 39~40 are used for testing. The number of samples in the two sets is 40, 320 and 16,560. (2) Cross-View, which means that actions from cameras go for training, and the action acquired by the camera labeled one is used for testing. Thirty-seven thousand nine hundred twenty from the whole dataset are used for training, and 18,960 actions are selected for testing. The NTU120RGB+D dataset is an extension of the NTU60RGB+D dataset. The combination of camera placement is 32, the action classification is increased to 120, the number of actors has reached 106, the number of action clips is increased to 114,480, and the number of sample joint points remains 25.

The reported 96.8% CV accuracy is based on the standard NTU60RGB+D protocol, where training and test views are entirely disjoint (training: cameras 2 and 3; testing: camera 1). Notably, the proposed method shows marked improvements in actions with large spatial variations or body-plane shifts, such as kicking, punching, and clapping. The attention modules (GAM and CAM) enhance spatial adaptability and feature robustness, enabling more accurate recognition under challenging view changes.

## Compare experiments between the designed model and others

To certify the comprehensive achievement of the modules here, experiments are compared among models, such as ST-GCN (*Yan, Xiong & Lin, 2018*), AS-GCN (*Liu et al., 2023*), Clips+CNN+MTLN (*Zhang et al., 2020*), and TCN (*Tian et al., 2024*; *Khan et al., 2024*). These models are more commonly listed among human skeleton data recognition. The dataset used in the experiment is NTU60RGB+D, and the evaluation indicators were CS and CV, serving as the evaluation criteria. The experimental results are drawn in Fig. 6.

The lightweight GCN designed here, based on the improved Ghost's attention mechanism, achieves better accuracy than other network structures on the NTU60RGB-D dataset. At the same time, the performance of the designed network structure is significantly more nuanced than that of other GNNs.

## Ablation study and module contribution analysis

To evaluate the individual contributions of the proposed components, we conducted an ablation study using the NTU60RGB+D dataset under CS and CV protocols; the results are shown in Table 2.

As shown in Table 2, introducing the Ghost module alone reduces the computational cost by over 60% compared to the ST-GCN baseline while improving recognition accuracy by nearly 5% (from 81.5% to 86.4%). Adding either the GAM or the CAM further enhances performance by 1.4–1.9%, confirming their role in improving spatial and channel-wise feature refinement. The complete model incorporating Ghost, GAM, and CAM achieves the highest accuracy (90.8% CS, 96.8% CV), validating the synergistic effect of all components. These findings demonstrate that each module makes a meaningful contribution to the model's overall efficiency and recognition performance.

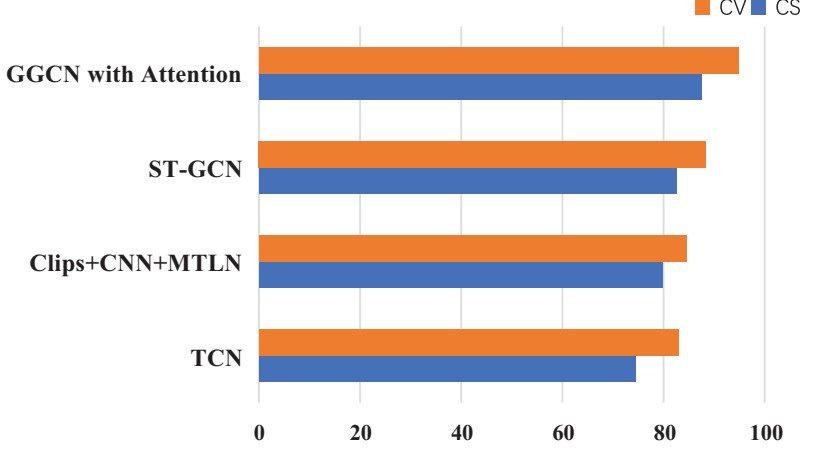

**Figure 6** Compare experiments between several models on NTU60RGB+D.

**Table 2 Ablation study on NTU60RGB+D dataset.**

| Model variant | GFLOPs | Params (M) | Accuracy (CS) | Accuracy (CV) |
|---|---|---|---|---|
| (1) Baseline ST-GCN (*Yan, Xiong & Lin, 2018*) | 16.2 | 3.1 | 81.5% | 88.3% |
| (2) + Ghost module only | 5.8 | 1.9 | 86.4% | 92.7% |
| (3) + Ghost + GAM only | 6.0 | 2.1 | 88.3% | 94.6% |
| (4) + Ghost + CAM only | 6.0 | 2.1 | 87.8% | 93.9% |
| (5) + Ghost + GAM + CAM (Full model) | 6.2 | 2.2 | 90.8% | 96.8% |

## Discussion

The findings of this study have several important implications for both research and practice in the field of human action recognition. First, the proposed Ghost-GCN architecture demonstrates that it is possible to achieve high-accuracy skeleton-based action recognition while significantly reducing computational cost and parameter size. This makes the model well-suited for deployment in real-time or resource-constrained environments, such as mobile health applications, wearable devices, and edge AI platforms like Raspberry Pi or NVIDIA Jetson Nano.

Second, the study highlights the effectiveness of attention mechanisms—specifically the GAM and CAM—in enhancing the representation of spatiotemporal dependencies across skeleton joints. These mechanisms improve generalization to challenging motion patterns, particularly under unseen viewpoints or actor variations, and could inform future designs of adaptive graph neural networks.

Third, by focusing on the context of online physical education, the proposed model addresses an urgent societal need for scalable, intelligent exercise monitoring systems in educational settings. Beyond education, the model holds promise for rehabilitation monitoring, elderly fall detection, and gesture control in AR/VR systems, where lightweight, interpretable, and low-latency models are essential.

Ultimately, this work lays the groundwork for future research that could integrate multimodal sensing data—including inertial measurement units (IMUs), audio, and electromyography (EMG)—to further enhance action understanding. Moreover, it encourages further exploration into model compression techniques, hardware-aware neural architecture search, and robust performance under noisy or incomplete skeleton inputs.

## CONCLUSION

This study presents a lightweight graph convolutional network with multi-attention mechanisms tailored for human skeleton-based action recognition in online physical education. The proposed model integrates an improved Ghost module and dual attention designs (GAM and CAM), achieving 96.8% accuracy under the cross-view setting and 90.8% under the cross-subject setting on the NTU60RGB+D dataset. Compared to the baseline ST-GCN, our method reduces floating-point operations by over 60% (from 16.2 to 6.2 GFLOPs) and parameter size by approximately 39% (from 3.1 million to 1.9 million), while substantially improving recognition accuracy. Beyond online physical education, the lightweight and accurate nature of the model suggests strong potential for broader applications, including rehabilitation monitoring, elderly movement analysis, and gesture-based interfaces in virtual or augmented reality environments. These scenarios similarly demand robust yet efficient skeleton-based recognition.

Future work will involve evaluating the model's real-time inference latency and deployment feasibility on edge devices such as Raspberry Pi or NVIDIA Jetson Nano. Additionally, integrating multimodal signals—such as wearable inertial measurement units (IMUs), ambient audio cues, or visual-textual annotations—could enhance context-aware recognition and extend the system's applicability to more complex activity environments.

### Funding
The author received no funding for this work.

### Competing Interests
The author declares that they have no competing interests.

### Author Contributions
- Yuhao You conceived and designed the experiments, performed the experiments, analyzed the data, performed the computation work, prepared figures and/or tables, authored or reviewed drafts of the article, and approved the final draft.

### Data Availability
The NTU60RGB+D dataset is available at Kaggle: https://www.kaggle.com/datasets/hungkhoi/skeleton-data-of-ntu-rgbd-60-dataset.

## Supplemental Information

Supplemental information for this article can be found online at http://dx.doi.org/10.7717/peerj-cs.3050#supplemental-information.

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
