# Peer review of "Lightweight graph convolutional network with multi-attention mechanisms for intelligent action recognition in online physical education"

_PeerJ Computer Science, doi:10.7717/peerj-cs.3050_

## Round 0.1 · original submission · Major Revisions

Dear authors
your manuscript has been reviewed by the experts and you will see that they have suggested major changes to be incoporated in your manuscript. Therfore, We adivse you to please update the manuscript in ligh of these suggested changes along with mine below and resubmit for further process

AE Comments

- The paper occasionally uses colloquial or non-academic language (e.g., “suiting the various example data”), which should be revised for clarity.

- Include a brief explanation of why CAM was selected to enhance anaerobic action expression specifically.

The paper introduces three modifications (Ghost module, GAM, CAM) but does not isolate their individual contributions. An ablation study should be conducted, and reported in a table comparing:

- Baseline GCN (e.g., ST-GCN): Ghost module only;Ghost + CAM only;Ghost + GAM only
- Full model with all components:Performance should be compared in terms of CS/CV accuracy, GFLOPs, and model size.

- Although GFLOPs are reported, parameter count and model file size are absent. For a model that claims to be “lightweight,” these are essential. Please provide parameter counts in millions and file size in MB for a fair comparison with AS-GCN, AGC-LSTM, etc.

- The CV accuracy is excellent (96.8%), but the setup lacks clarity. Please clarify: Whether test views are disjoint from training views. If so, what types of motions benefit most from the dual attention mechanism under unseen views?

- The conclusion section merely restates the model structure and claims. Consider expanding it to include: Quantitative gains (e.g., “FLOPs reduced by 60% compared to ST-GCN”). Discussion on real-time deployment potential (e.g., Jetson Nano, mobile ARM devices)

- Consider including inference time in milliseconds/frame as a practical metric.

Reviewer 1 ·

Basic reporting

The paper proposes a lightweight GCN enhanced with an improved Ghost module and two attention mechanisms (GAM and CAM), which is an interesting and practical design. However, the novelty relative to existing works such as AGC-LSTM and AS-GCN should be more clearly emphasized—e.g., how your dual attention mechanism synergistically improves over previous two-stream designs.
Figure 2 is informative, but its flow is difficult to follow due to overlapping arrows and repeated components. Consider redrawing it with labeled input/output dimensions and clear sequential arrows. Additionally, it would help to summarize the complete pipeline in one paragraph in Section 3 before breaking down into modules.
The paper highlights reduced GFLOPs and high accuracy, which supports the efficiency claim. However, please provide model size in MB or parameter count (e.g., millions of parameters) for fairer comparison with other lightweight methods.

Experimental design

Currently, only full model comparisons are provided. To better support claims, include ablation experiments isolating: (a) Ghost module alone, (b) only GAM or CAM, (c) no attention, to quantify each component’s contribution to performance and efficiency.
The paper states 96.8% accuracy on CV, which is excellent. Please clarify whether the views in training and test are disjoint, and if so, explain what types of motions are most improved by your method under unseen camera angles.
While the model is positioned for online physical education, its utility for rehabilitation, elderly movement monitoring, or VR interfaces could be mentioned in the Discussion section to broaden the impact scope.
The current conclusion is brief and largely reiterative. Strengthen this section by:
• Summarizing empirical gains (e.g., % reduction in FLOPs vs. ST-GCN).
• Recommending future experiments, such as real-time latency tests or hardware deployment scenarios (e.g., Raspberry Pi, Jetson Nano).
• Exploring integration with multimodal data (e.g., wearable IMUs or audio cues).
In Section 3.3.1, the mechanism by which the data-driven graph matrix (A′) and attention matrix (B) are jointly updated during training is mentioned but not fully explained. Please clarify whether this is learned end-to-end, and whether A′ is fixed post-initialization or updated per epoch.
There are several instances of inconsistent or undefined variables:
o “f′ ∈ Rc×k×k×m” appears without defining m clearly.
o The difference between matrices A, A′, and B should be better structured and summarized in a table or boxed formula summary.
o Symbols like “α is set to 0.001” are introduced without prior definition of where they apply (e.g., normalization?).
The manuscript overall is readable but contains numerous typographic issues. For example:
o “literally voluntary to generate relatively spare feature maps…” is unclear and unidiomatic.
o “so as to suiting the various example data…” should be corrected to “so as to suit…”
Please perform a thorough grammar check and revise non-standard English expressions, especially in Sections 3.2 and 3.3.

Validity of the findings

Table 1 includes GFLOPs but lacks standard deviation or variance measures. Please add error bars or ranges to accuracies if available, or state if all runs are deterministic. Also, visually emphasize (bold or underline) your best-performing metrics for clarity.

Reviewer 2 ·

Basic reporting

The idea presented in this paper “Lightweight graph convolutional network with multi-attention mechanisms for intelligent action recognition in online physical education” is good. However, the manuscript needed to be improved by keeping in view following suggestions.

1: The abstract needed to provide a more precise synthesis of the work done and findings is a structured flow covering background, research gap / problem, proposed approach, experimental findings.

2: In the abstract author stated “To address issues such as high computational costs and insufficient feature extraction capabilities, this study proposes a lightweight Graph Convolutional Network (GCN) model with multi-attention mechanisms for human skeletal action recognition.”

How the proposed approach is has computational cost? As it is one of the problems mentioned. How the computational cost is calculated in this study for proposed and existing techniques.

3: line 78-79 keeping in reference point 2. It is suggested to added a comparison for the computational cost as well.

4: Add few lines at the end of introduction section that details how this manuscript is organized in different sections.

5: The related work needed to be extended and improved by adding more recent studies

Experimental design

6: Section 4.1, add missing citation for NTU60RGB+D dataset.

7: Section 4.2, the discussion of the experimental results needed to be extended and discussed in more detail.

Validity of the findings

8: Add a paragraph and discuss the comparison with existing studies.
9: Add a section to discuss the implications of work done in this study.
10: Add one more paragraph at the end of conclusion that highlights the limitations of work done in this study and also list potential future research directions.

---

## Round 0.2 · accepted · Accept

Thank you for your re-submission and updation in light of expert's comments. I'm pleased to inform you that the experts are satisfied with your revised updated version and they recommend acceptance. So I endorse their decision. Thank you for your fine contribution

Reviewer 1 ·

Basic reporting

I have thoroughly evaluated the revised manuscript and the authors’ detailed responses to the reviewer comments. The authors have addressed all concerns thoughtfully and effectively. The revisions have notably enhanced the clarity, coherence, and overall quality of the manuscript.

I am satisfied with the improvements made and find the paper suitable for publication. I recommend acceptance of the paper in its current form.

Experimental design

Good

Validity of the findings

Good

Additional comments

I have thoroughly evaluated the revised manuscript and the authors’ detailed responses to the reviewer comments. The authors have addressed all concerns thoughtfully and effectively. The revisions have notably enhanced the clarity, coherence, and overall quality of the manuscript.

I am satisfied with the improvements made and find the paper suitable for publication. I recommend acceptance of the paper in its current form.

Reviewer 2 ·

Basic reporting

no comment

Experimental design

no comment

Validity of the findings

no comment